# Study on the Actuation Properties of Polyurethane Fiber Membranes Filled with PEG-SWNTs Dielectric Microcapsules

**DOI:** 10.3390/membranes12101026

**Published:** 2022-10-21

**Authors:** Gang Lu, Changgeng Shuai, Yinsong Liu, Xue Yang

**Affiliations:** 1Institute of Noise and Vibration, Naval University of Engineering, Wuhan 430033, China; 2Key Laboratory of Ship Vibration and Noise, Wuhan 430033, China

**Keywords:** polyurethane dielectric elastomer, dielectric microcapsule, coaxial spinning, fiber membrane, actuation properties

## Abstract

Polyurethane dielectric elastomer (PUDE), a typical representative of emerging intelligent materials, has advantages, such as good elasticity and flexibility, fast response speed, high electromechanical conversion efficiency, and strong environmental tolerance. It has promising applications in underwater bionic actuators, but its electromechanical properties should be improved further. In this context, the design of polyethylene glycol (PEG) single-walled carbon nanotube (SWNTs) dielectric microcapsules was adopted to balance the problem of contradictions, which conventional dielectric modification methods face between comprehensive properties (e.g., dielectric properties and modulus). Moreover, the dielectric microcapsule was evenly filled into the polyurethane fiber by coaxial spinning technology to enhance the actuation performance and instability of the electrical breakdown threshold of conventional polyurethane dielectric modification. It was revealed that the dielectric microcapsules were oriented in the polyurethane fiber, and the actuation performance of the composite fiber membrane was significantly better than that of the polyurethane fiber membrane filled with SWNTs, thus confirming that the filling design of the dielectric microcapsules in polyurethane fiber could have certain technical advantages. On that basis, this study provides a novel idea for the dielectric modification of polyurethane.

## 1. Introduction

A dielectric elastomer (DE) refers is a novel intelligent material that is capable of producing large shape and size deformation under an external electric field [1,2,3,4]. Compared with conventional intelligent materials (e.g., shape memory alloys (SMAs), piezoelectric ceramic (PZT), and magnetostrictive material (MSM), DE materials have shown advantages of a fast response, large electrical distortion, good elasticity, high efficiency of electromechanical transformation, etc. They are considered promising for applications in new research fields of intelligent materials (e.g., biomimetic mechanical design) [5,6,7,8].

The control design of actuating elements and devices is conducted using the mechanical energy transformed by DE materials based on a high-voltage stimulation. Research on land soft robots, flexible sensing actuators, and artificial muscle has been extensively conducted [9,10], while research on underwater bionic platforms is still preliminary. This is because the underwater environment is more complicated than that on land. At present, DE-actuating materials (e.g., acrylic resin and silicone rubber) face problems of high temperature and humidity sensitivity and modulus self-enhancement effects [11,12]. Moreover, these DE composites modified by conventional dielectric techniques are accompanied by other problems, including increasing modulus, low dielectric sensitivity, and a low electric breakdown threshold that arises from the agglomeration of small particle dielectric fillers [13,14]. The main cause of agglomeration is excessive specific surface area. In addition, the distance between particles is very short, and the van der Waals forces between particles are far greater than their own gravity, which often leads to attraction and agglomeration. DE-actuating materials are difficult to effectively apply to the future design of flexible actuators and devices of an underwater bionic platform. Therefore, it is revealed that we should enhance the dielectric properties of DE, balance the increase in material modulus, and make the limited dielectric filler as evenly dispersed as possible in DE to fully exploit its modification function. Only then can we offer basic technical support for the development of a DE-actuating element and device exhibiting a stable actuating performance.

Moreover, PUDE has advantages of a high mechanical and electrical conversion speed as a typical flexible material, has a flexible molecular structure, strong mechanical properties, and a strong tolerance to the environment [15,16,17,18]. In addition, the solvent dissolution characteristics cause the processing methods of linear thermoplastic PUDE to be diverse [19]. On the other hand, the existing literature found that the balance between dielectric properties and the modulus generally adopts a method of successively adding dielectric filler and plasticizer to DE materials [20,21,22], but this scheme inevitably leads to a decrease in the comprehensive properties of dielectric composites.

Accordingly, dielectric microcapsules have been proposed, with a dielectric material as the core and with a flexible material as the wall structure, to balance the contradiction between dielectric properties and the modulus. Furthermore, coaxial spinning technology [23] was introduced to account for the small-scale particle agglomeration caused by the loss of stirring force in conventional blending by directional constraint molding. The dielectric microcapsules were filled into the polyurethane fibers, and the electromechanical properties of the polyurethane composite fiber membrane were investigated.

## 2. Experimental Techniques

### 2.1. Materials

Distilled water (DW, AR, Xiangjiutong Biotechnology Co., Ltd., Wuhan, China), polyethylene glycol-8000 (PEG-8000, MW: 8000, Lianji Chemical Co., Ltd., Shanghai, China), single-walled carbon nanotubes (SWNTs, OD < 2 nm, China Science Times Nanocenter, Chengdu, China), sodium chloride (NaCl, CP, Xiangjiutong Biotechnology Co., Ltd., Wuhan, China), ether (ET, CP, Xiangjiutong Biotechnology Co., Ltd., Wuhan, China), *N*,*N*-dimethylformamide (DMF, CP, Xiangjiutong Biotechnology Co., Ltd., Wuhan, China), polyether MDI polyurethane prepolymer (MDI-PUP, CP, Polymer Rubber Products Co., Ltd., Dongguang, China), polyether polyol additive (PA, CP, Dongxu Chemical Industry Manufacturing Co., Ltd., Heshan, China), and 1,4-butanediol (BDO, CP, Tianmao Chemical Co., Ltd., Xi’an, China) were used in this research.

### 2.2. Devices

Table 1 lists the main devices involved in this study.

### 2.3. Synthesis of PEG-SWNTs Dielectric Microcapsules

#### 2.3.1. Formulation Design

Table 2 presents the formulation design of dielectric microcapsules.

#### 2.3.2. Design Route

Figure 1 illustrates the synthesis of PEG-SWNTs.

#### 2.3.3. Specific Steps

(1) First, the PEG-8000 solution and SWNTs suspension were prepared in accordance with the formula listed in Table 1, respectively. Subsequently, the SWNTs suspension was slowly poured into the PEG-8000 solution at 65 °C and stirred evenly. (2) The obtained mixture was placed in an ultrasonic environment at 65 °C for continuous oscillation for 12 h. Next, the mixture was transferred to the environment at 70 °C for continuous stirring at 1200 rpm. Moreover, the coagulant sodium chloride solution was slowly introduced to the mixture. (3) Due to the continuous dropping of coagulant, the solubility of the PEG in water tended to decrease. Once the critical state was attained, the supersaturated PEG began to settle on the condensed core of SWNTs until the complete dielectric microcapsule was formed. Lastly, the collected dielectric microcapsule was dried in a 30 °C oven for 24 h for standby.

### 2.4. Preparation of Polyurethane Composite Fiber Membrane Filled with Dielectric Microcapsules

#### 2.4.1. Synthesis of PUDE

The preparation of the polyurethane dielectric elastomer is presented in Figure 2. First, MDI-PUP, PA, and BDO were prepared at mass ratios of 100 phr, 30 phr, and 12.8 phr, respectively. Subsequently, the above components were mixed, vacuumized, poured and matured in accordance with the procedures shown in the figure. Lastly, PUDE was obtained for standby.

#### 2.4.2. Coaxial Spinning Technology and Formula Setting of the Membrane

(1) Coaxial spinning technology. The preparation of the polyurethane fiber membrane filled with dielectric microcapsules requires a coaxial spinning device, and its forming principle is illustrated in Figure 3. As depicted in Figure 3, the device primarily consists of three parts: the fiber generation device, fiber collection device, and control center. The first part largely contains a coaxial dual-channel spinning head, in which the inner channel was connected with the ET dispersed with dielectric microcapsules; the outer channel was connected with the DMF dissolved with PUDE. As a result, with the continuous loading of high voltage, the dual-channel solution pushed out of the spinning head in a Taylor cone flow. The fiber was then continuously collected on the roller wrapped in tin foil and connected with negative voltage. Lastly, polyurethane composite fiber membranes were formed. During the above process, the solution concentration, voltage, spinning head scanning speed, collection roller speed, and other environmental factors (e.g., temperature and humidity) would be regulated by the control center. Subsequently, a polyurethane composite fiber membrane with a good morphology could be produced.

(2) Formulation design. To compare the electromechanical properties of the polyurethane composite fiber membrane filled with SWNTs or PEG-SWNTs (DMC), the formula in Table 3 was designed, and the content of single-wall carbon nanotubes in the polyurethane composite fiber membrane of the two systems was essentially the same, thus ensuring that the comparative experimental research is scientific.

#### 2.4.3. Specific Steps

(1) First, the lighting was turned on in the spinning box, and the prepared spinning solution of channel 1 and channel 2 was injected into the shaft and shaft sleeve channels, respectively, in accordance with the formula shown in Table 3. Subsequently, the initial position of the spinning head was adjusted. Moreover, the tin foil was fixed with an appropriate width on the collection roller. After all preparation steps were completed, the safety door was closed. (2) The control center was opened, and the distance between the spinning head and collecting drum, the glue pushing speed, the voltage value, the drum speed, the scanning point, and distance were adjusted. Next, the spinning parameters were slowly adjusted and determined based on the shape of the Taylor cone flow in the spinning head. (3) After preparing the fiber membrane, all buttons in the control center were first closed in sequence, and then the safety door was opened. Next, the tin foil was carefully removed from the collection drum and transferred to an oven with a temperature of 30 °C for 12 h of drying. Lastly, the polyurethane fiber membrane was removed from the tin foil, and the characterization and performance testing were conducted. The polyurethane dielectric fiber membrane (DEM) without any dielectric filler, which was the control group of the polyurethane composite fiber membrane of the two systems, could be prepared using electrospinning, i.e., a conventional method, but this was not elucidated in this study.

### 2.5. Characterization and Testing

#### 2.5.1. Characterization of Dielectric Microcapsules

Table 4 presents the characterization items of dielectric microcapsules.

#### 2.5.2. Characterization and Testing of Polyurethane Composite Fiber Membrane

(1) Characterization of the polyurethane fiber membrane. First, the micromorphology of the cross section of the fiber membrane was characterized under a scanning electron microscope to observe the dispersion of the dielectric filler in the fiber bundle. Table 5 lists the equipment information.

(2) Dielectric sensitivity coefficient test. In general, the dielectric sensitivity coefficient β, i.e., electromechanical sensitivity factor, is correlated with dielectric constant and elastic modulus, as expressed in Equation (1).
(1)$$\beta =\frac{\varepsilon_{r}}{Y}$$

It was revealed that while enhancing the dielectric properties of dielectric materials, it was also necessary to balance the elastic modulus of materials. (1) Dielectric constant test: The dielectric properties of the samples were measured with a 6632-1s dielectric constant tester from the Teng Skye company. The test frequency ranged from 10 Hz to 500 Hz, and the sample diameter was 10 mm. (2) Elastic modulus test: The sample was cut into several standard dumbbell-shaped samples. Using a strain rate of 200 mm/min, the elastic modulus was calculated from the slope of the initial part of the stress–strain curve (deformation less than 5%). The median of five parallel test values was taken as the final result.

(3) Electric breakdown voltage and electric deformation test: (1) The polyurethane composite fiber membrane was fixed on an epoxy resin insulation frame, and then the circular flexible electrode of perfluoropolyether conductive grease was evenly coated with a diameter of 15 mm on the front and back of the membrane with a cotton stick. Next, the copper foil was bonded to lead out the positive and negative electrodes, respectively, to obtain the polyurethane composite fiber membrane actuating unit. (2) The test circuit was connected, the focal length of the high-speed camera was adjusted and fixed in front of the actuating unit, and the electric deformation test platform of the actuating unit was built, as depicted in Figure 4. (3) During the test, the excitation voltage was gradually increased from 0 kV until the electric breakdown of the fiber membrane. Photos of the deformation of the fiber membrane actuating unit and the corresponding voltage value in this process were recorded. (4) The power supply was turned off, and the test data were sorted and analyzed. The area was obtained before and after electro deformation using pixel lattice processing in Photoshop software. Strain S_A_ caused by the electric deformation of the actuating unit can be calculated by Equation (2):(2)$$S_{A}=\frac{S_{2}-S_{1}}{S_{1}}\times 100\%$$
where S_1_ and S_2_ represent the areas of the coated electrode area before and after the deformation of the actuating unit, respectively.

The collected images were analyzed by software, i.e., the area strain of the polyurethane composite fiber membrane actuating unit was obtained by calculating the pixel change in the flexible electrode area under different voltages.

## 3. Results and Discussion

### 3.1. Dielectric Microcapsule

#### 3.1.1. Infrared Spectrum Analysis

As depicted in Figure 5, the stretching vibration peak of OH appeared at 3400 cm^−1^. In the figure, 1375 cm^−1^, 1250 cm^−1^, and 870 cm^−1^ represent the positions of the characteristic functional groups of CH_2_. The positions of 1095 cm^−1^ and 950 cm^−1^ represent the characteristic peaks of C–O–C. The appearance of the above characteristic functional groups confirmed the existence of PEG. Moreover, the characteristic absorption peaks of SWNTs were generally at 1558 cm^−1^~1583 cm^−1^ and 1180 cm^−1^~1190 cm^−1^, while there were no significant characteristic absorption peaks, as shown in Figure 5. This result further reveals that the SWNTs were wrapped by PEG.

#### 3.1.2. Particle Size Distribution

Figure 6 presents the particle size distribution curve of the dielectric microcapsule of PEG 8000-SWNTs. As depicted in Figure 6, the particle size distribution ranged from 1.0 μm to 6.0 μm, and the average particle size was concentrated at 3.5 μm.

### 3.2. Polyurethane Composite Fiber Membranes

#### 3.2.1. Fiber Morphology and Arrangement Trend of Dielectric Microcapsules in Fibers

As depicted in Figure 7a, the continuity of fibers in the polyurethane composite fiber membrane was high, and no significant agglomeration was identified. One place in the figure was selected randomly for accelerated voltage transmission to generate Figure 7b. As revealed by the figure, PEG 8000-SWNTs dielectric microcapsules and SWNTs dielectric fillers were oriented in the fiber bundle, to be consistent with the design idea of this study, which can be used to measure the dielectric and electrostrictive properties exhibited by the polyurethane composite fiber membrane in the next step.

#### 3.2.2. Dielectric Properties

Table 6 lists the dielectric property data of polyurethane fiber membrane filled with dielectric microcapsules of SWNTs or PEG-SWNTs with the same formula.

To more effectively compare the data in the table with their corresponding change trend, Figure 8 is drawn from the data in the table.

The upper left figure illustrates the comparison diagram of the loss factors of polyurethane fiber membranes of the two systems at 10 Hz and 100 Hz. As indicated by the information in the figure, the dielectric loss factor of SWNTs-DEM increased significantly compared to the DMC-DEM at 10 Hz, while the loss factor advantage of the DMC-DEM at 100 Hz was not significant. It was therefore revealed that the flexible capsule wall design of dielectric microcapsules could promote a reduced loss factor of SWNTs at low frequencies. When the frequency increased, the effect tended to decrease.

The right figure compares the dielectric sensitivity factors of polyurethane fiber membranes of the two systems at 10 Hz and 100 Hz. As depicted in this figure, at low frequencies, the dielectric sensitivity factor of the DMC-DEM was significantly higher than that of the SWNTs-DEM as the filling fraction of dielectric filler in the fiber increased. Combined with the Young’s modulus of the fiber membranes of the two systems in Table 6, on the premise of the design of the dielectric filler with the same formula, the design of the dielectric microcapsule did not affect the dielectric constant of the fiber membrane. Its flexible capsule wall could significantly reduce the Young’s modulus of the fiber membrane, which was found to increase the dielectric sensitivity of the DMC-DEM. Moreover, since the dielectric constant of the composites exhibited a large frequency dependence and decreased with the increased frequency, the dielectric sensitivity factor of the DMC-DEM at 100 Hz did not show any significant advantage over the SWNTs-DEM.

#### 3.2.3. Electrical Deformation Properties

Table 7 presents the electrical deformation properties of polyurethane composite fiber membranes of two systems, which consist of electrical breakdown properties and electrical deformation properties.

To clearly illustrate the change trend and comparison of polyurethane composite fiber membranes of the two systems, Figure 9 is drawn in accordance with the data listed in the table below.

The upper left figure presents the breakdown voltage changes of SWNTs-DEM and DMC-DEM. The figure shows that as the filling fraction of dielectric filler in the matrix material increased, the breakdown voltage of the fiber membrane of the two systems tended to decrease. The difference was that the breakdown voltage of the DMC-DEM system material was higher than that of SWNTs-DEM, and the decrease in the amplitude of the breakdown voltage of the former also decreased with the increase in dielectric filler. The above difference suggests that the design of the dielectric microcapsule could effectively improve the dispersion behavior of dielectric filler in the matrix, and its flexible capsule wall largely separated the physical agglomeration between SWNTs, thus hindering the formation of the conductive path.

The left figure above reveals that the loading voltage of the actuating fiber membrane of the two systems could be adjusted to be lower than 30 kV. On that basis, the actuating performance of the actuating unit of the polyurethane composite fiber membrane could be investigated. Accordingly, 10 kV, 20 kV, and 30 kV were selected as the control variables in this section. As depicted in the right figure, as the loading voltage increased, the DMC-DEM system fiber membrane exhibited better electrical deformation properties than those of the SWNTs-DEM system, which is consistent with the law of the dielectric sensitivity of the polyurethane fiber membrane of the two systems.

## 4. Conclusions

In this study, dielectric microcapsules, with PEG-8000 as the flexible capsule wall and SWNTs as the capsule core, were developed using a comparative research method. Subsequently, coaxial spinning technology was employed to evenly fill the dielectric filler into the polyurethane fiber, and polyurethane composite fiber membranes filled with dielectric microcapsules and SWNTs were prepared to solve the performance contradictions of the conventional dielectric modification methods of PUDE. Lastly, the following conclusions were drawn through a series of characterization and tests:

(1) The design of a dielectric microcapsule can effectively reduce the Young’s modulus of the polyurethane composite fiber membrane and increase its dielectric sensitivity by leaving its dielectric constant unchanged. The above method provides a new method to improve the dielectric sensitivity of dielectric composites.

(2) The polyurethane fiber membrane filled with dielectric microcapsules in the same formula exhibited better electrical deformation properties than that filled with SWNTs at varying degrees. This shows that the plastic wall of the dielectric microcapsule can effectively stabilize the electric breakdown threshold of the membrane material and promote a reduction in the loss factor of SWNTs at low frequencies.

(3) The directional constraint molding method of coaxial spinning technology can effectively facilitate the agglomeration of micro- and nanodielectric particles in the matrix material caused by conventional physical blending. The dispersion of dielectric filler in the matrix material is effectively improved, which is conducive to enhancing the electrical breakdown properties of dielectric composites. Furthermore, this method can provide a reference for the uniform dispersion of multiphase micro- and nanofillers in composite research.

## Figures and Tables

**Figure 1 membranes-12-01026-f001:**
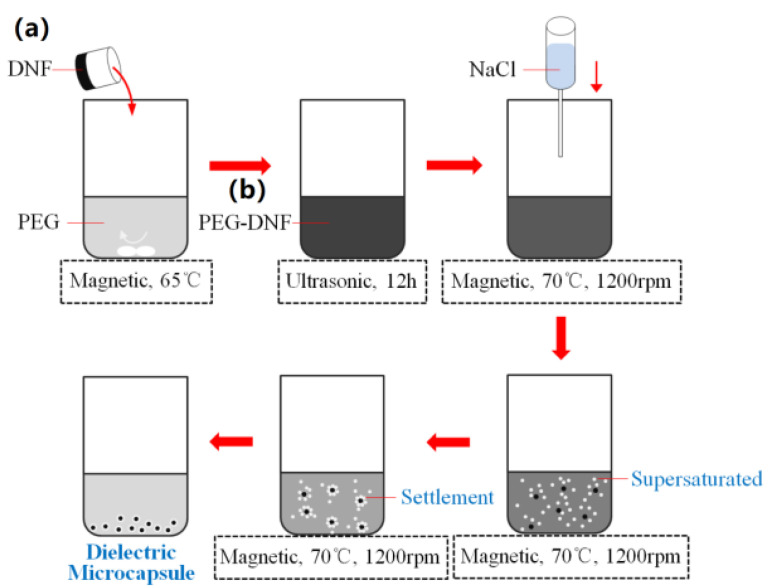
Design route of PEG-SWNTs dielectric microcapsules: (**a**) DNF: Dielectric Nanofiller; (**b**) PEG-DNF: PEG-Dielectric Nanofiller.

**Figure 2 membranes-12-01026-f002:**
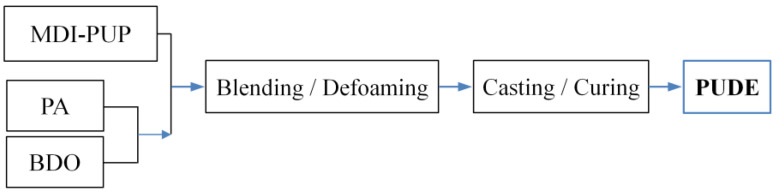
Preparation Flow of PUDE.

**Figure 3 membranes-12-01026-f003:**
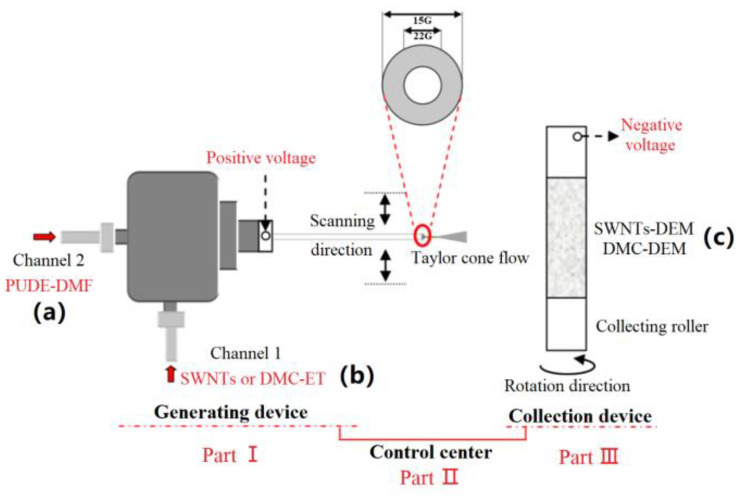
Schematic diagram of polyurethane fiber membrane forming based on coaxial spinning. (**a**) PUDE-DMF is a PUDE solution dissolved in N, N-dimethylformamide; (**b**) SWNTs or DMC-ET are SWNTs or the dielectric microcapsules dispersed in ether; (**c**) SWNTs-DEM or DMC-DEM are the polyurethane composite fiber membranes filled with SWNTs or dielectric microcapsules, respectively.

**Figure 4 membranes-12-01026-f004:**
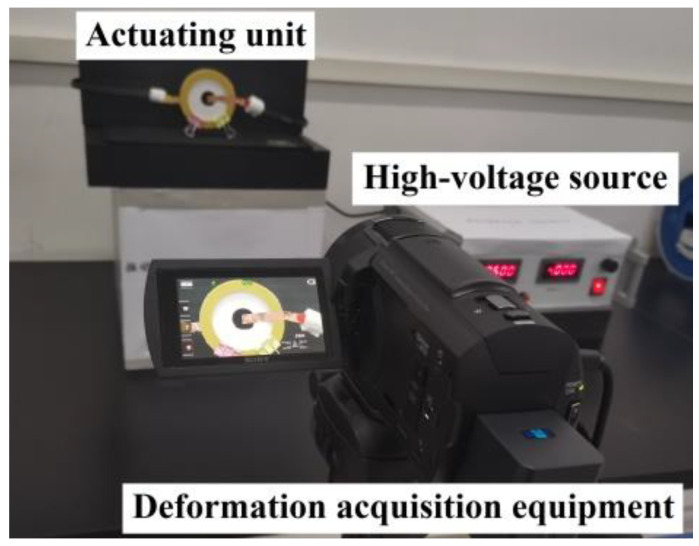
Electrostrain experimental platform.

**Figure 5 membranes-12-01026-f005:**
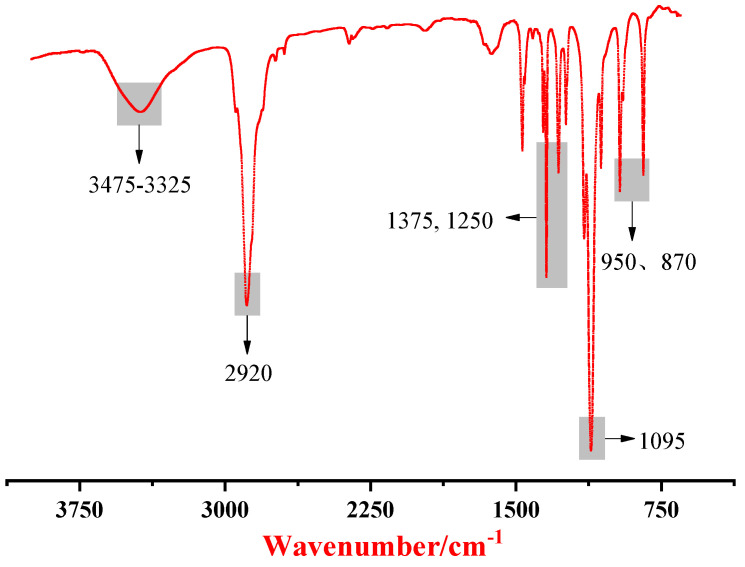
Infrared spectra of dielectric microcapsules.

**Figure 6 membranes-12-01026-f006:**
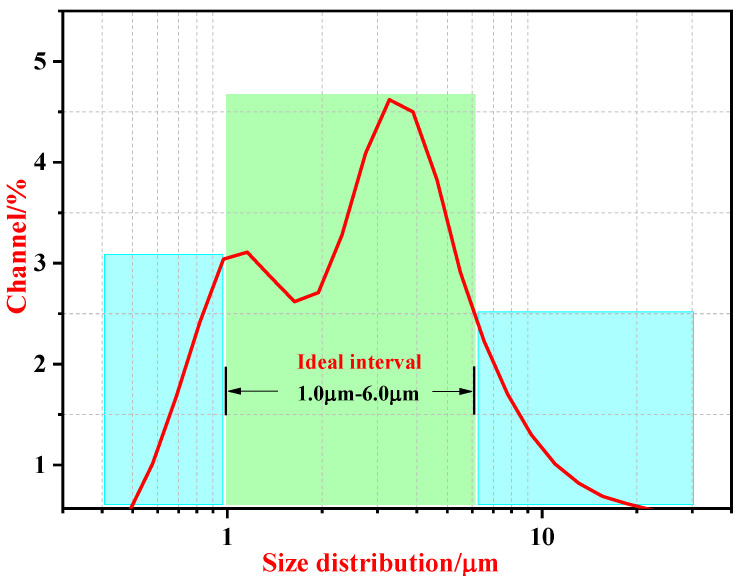
Particle size distribution of PEG 8000-SWNTs dielectric microcapsules.

**Figure 7 membranes-12-01026-f007:**
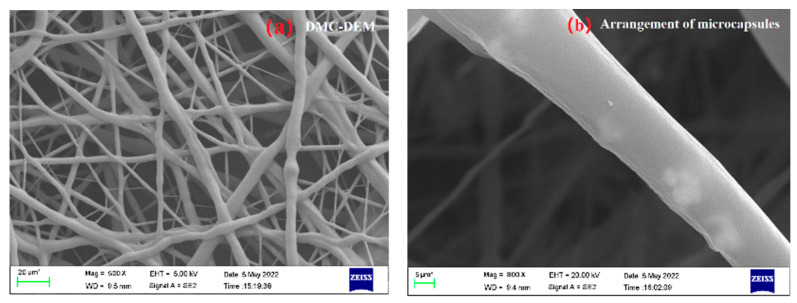
Arrangement trend of dielectric microcapsules in fibers. (**a**) Fiber morphology; (**b**) arrangement trend of dielectric microcapsules in fibers.

**Figure 8 membranes-12-01026-f008:**
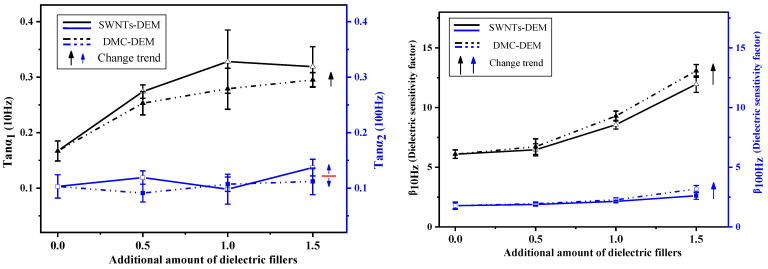
Comparison and change trend of dielectric properties of polyurethane fiber membrane.

**Figure 9 membranes-12-01026-f009:**
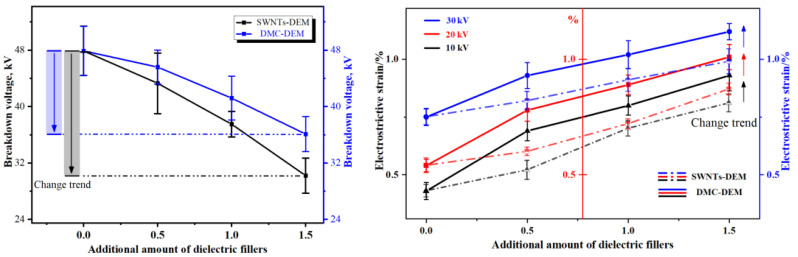
Electrical deformation properties of series polyurethane composite fiber membranes.

**Table 1 membranes-12-01026-t001:** The main devices involved in the experimental process.

S/N	Instrument	Manufacturer	City
1	Constant temperature magnetic stirrer	Meiyingpu Instrument Manufacturing Co., Ltd.	Shanghai, China
2	Circulating water vacuum pump	Great Wall Technology Industry and Trade Co., Ltd.	Zhengzhou, China
3	Vacuum drying oven	Zhuocheng Technology Co., Ltd.	Zhengzhou, China
4	Electrospinning apparatus	Tongli Co., Ltd.	Shenzhen, China

**Table 2 membranes-12-01026-t002:** Formulation design of dielectric microcapsules.

Items	PEG-8000 Solution/phr		SWNTs Suspension/phr		NaCl Solution/phr	
	PEG-8000	DW	SWNTs	DW	NaCl	DW
1	5	5	0.20	10	10	30
2	5	5	0.40	10	10	30
3	5	5	0.60	10	10	30

Notes: phr is the mass fraction.

**Table 3 membranes-12-01026-t003:** Formulas of the polyurethane composite fiber membranes.

Items	Channel 2		Channel 1		
	PUDE	DMF	SWNYs	DMC	ET
DEM	20	80	/	/	/
SWNTs-DEM_1_			0.1		50
SWNTs-DEM_2_			0.2		50
SWNTs-DEM_3_			0.3		50
DMC-DEM_1_			/	2.6	50
DMC-DEM_2_				2.7	50
DMC-DEM_3_				2.8	50

The thickness of the polyurethane composite membrane should be about 1 mm.

**Table 4 membranes-12-01026-t004:** Characterization items of dielectric microcapsules.

Items	Model	Remarks
Infrared spectrum analysis	TENSOR27	Test range is 600 cm^−1^–3600 cm^−1^
Morphology analysis	Su8010	Maximum magnification 2,000,000 times
Particle size analysis	QL-1076	0.1 μm–600 μm

**Table 5 membranes-12-01026-t005:** Device information.

Device	Model	Remarks
Electron microscope	Zeiss Merlin	Accelerating voltage was 20 kV

**Table 6 membranes-12-01026-t006:** Dielectric properties of polyurethane fiber membrane.

Samples	10 Hz		100 Hz		Y/MPa	β_10 Hz_	β_100 Hz_
	ε_1_″	Tanα_1_	ε_2_″	Tanα_2_			
DEM	14.079	0.167	4.104	0.103	2.31 ± 0.06	6.095	1.777
SWNTs-DEM_1_	17.314	0.274	5.037	0.119	2.68 ± 0.06	6.460	1.879
SWNTs-DEM_2_	26.975	0.358	6.729	0.098	3.15 ± 0.06	8.563	2.136
SWNTs-DEM_3_	41.502	0.311	9.067	0.137	3.47 ± 0.06	11.960	2.613
DMC-DEM_1_	16.325	0.263	4.731	0.091	2.43 ± 0.06	6.718	1.947
DMC-DEM_2_	26.471	0.249	6.467	0.107	2.85 ± 0.06	9.288	2.269
DMC-DEM_3_	39.616	0.295	9.635	0.112	3.03 ± 0.06	13.074	3.180

DEM_1–3_: Represents polyurethane composite fiber membrane filled with different content of dielectric filler.

**Table 7 membranes-12-01026-t007:** Electrostrain data of a series of polyurethane composite membranes.

Samples	Breakdown Voltage/kV	Electrostrictive Strain/%		
		10 kV	20 kV	30 kV
DEM	47.9	0.43	0.54	0.75
SWNTs-DEM_1_	44.7	0.52	0.60	0.82
SWNTs-DEM_2_	40.9	0.70	0.72	0.91
SWNTs-DEM_3_	37.5	0.81	0.87	0.99
DMC-DEM_1_	45.8	0.69	0.78	0.93
DMC-DEM_2_	42.2	0.80	0.89	1.02
DMC-DEM_3_	38.8	0.93	1.01	1.12

DEM_1–3_: Represents polyurethane composite fiber membrane filled with different content of dielectric filler.

## Data Availability

The data presented in this study are available on request from the corresponding author.

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
