# Peer review of "Study on the Actuation Properties of Polyurethane Fiber Membranes Filled with PEG-SWNTs Dielectric Microcapsules"

_membranes, 2022, doi:10.3390/membranes12101026_

Round 1

Reviewer 1 Report

Report to Membranes (MDPI Journal)

Article: Study on the Actuation Properties of Polyurethane Fiber Membranes Filled with PEG-SWNTs Dielectric Microcapsules

The paper under consideration reports the preparation of polyurethane composite fiber membrane filled with dielectric microcapsules. The synthesis of PEG-SWNTs dielectric microcapsules with PEG8000 as the flexible capsule wall and SWNTs as the capsule core, and the preparation of polyurethane dielectric elastomer were presented as well as their structural and morphologic characterization. The electrospinning technique was employed to fill the dielectric microcapsules in the polyurethane fiber, and the obtained polyurethane composite fiber membranes. This study contributes to an innovative idea for the dielectric modification of polyurethane.

The work is well organized and well presented. Data and methods are reported with rigor. Data analysis is performed correctly. Therefore, I have only minor revisions to request with respect to this paper.

1-     In the title, where it is written…”Mocrocapsules”…should be…microcapsules;

2-     In the introduction section the authors wrote, “…DE composites modified by conventional dielectric…” (page 1, lines 42-43). However, it would be interesting to better clarify these conventional processes and their particularities, and make reference to recent works.

3-     In the introduction section the authors wrote, “coaxial spinning technology was introduced to make up for the small-scale particle agglomeration” (page 2, lines 62-63). The technique used, Electrospinning, and the parameters involved in the process should be more extensible described. A reference should be introduced, as example: (Greiner,A.; Wendorff, J. H. Electrospinning: A Fascinating Method for the Preparation of Ultrathin Fibers. Angewandte Chemie International Edition 2007, 46,5670-5703)…or (Agarwal, S.; Greiner, A.; Wendorff, J. H. Functional materials by electrospinning of polymers. Prog Polym Sci 2013, 38, 963-991).

4-     Where it is written “mass ratios of 100thr,”…should be mass ratios of 100 phr.

5-     In the experimental section the authors wrote,“…the safety door was opeded.”…they wanted to write, opened.

6-     It is important to check the numbering of figures throughout the manuscript. Where it is written “…characteristic absorption peaks in Fig. 4”….should be:..in figure 5. (page 7, line 203).

7-     The microcapsules particle size characterization, before the incorporation into the polymeric matrix, was performed by laser. Other techniques, such as Dynamic Light Scattering (DLS), or Scanning Electron Microscopy (SEM) could be used. What was the reason for choosing Laser particle sizer/QL-1076 for such measurements?

8-     Where it is wrote “…As depicted in Fig. 5, the particle size”….should be:..figure 6. (page 7, line 207).

9-     Where is written “…partcle”….should be:..particle. (page 7, line 210).

10-  In table 4, where it is wrote “Divece”…should be device.

11-  Why the structural characterization of microcapsules and membranes was not performed by XRD? You should better substantiate the following sentence... “dielectric microcapsules and SWNTs dielectric fillers were oriented in the fiber bundle”. (page 7, lines 216-217). In my opinion, through SEM analysis, it is possible to observe a random orientation of the fibers, with the microcapsules having a uniform distribution inside them.

12-  Throughout the manuscript several observations are well explained, see for example at page 8 “It was therefore revealed that the flexible capsule wall design of dielectric microcapsules could promote the reduction of the loss factor of SWNTs at low frequencies.” Please use this observation in the conclusion to expand the view of the reader into possible applications of these new materials.

Reviewer 2 Report

The paper “Study on the Actuation Properties of Polyurethane Fiber Membranes Filled with PEG-SWNTs Dielectric Mocrocapsules” membranes-1967528 deals with the modulation of dielectric properties of PU using coaxial spinning. The manuscript presents extensive characterizations through IR, SEM and particle analysis. I think the results are interesting and don't have concerns about the characterizations. On the other hand, the paper as it is presented much more dedicated to the material characterization, but the results concerning the electromechanical coupling must be improved. Moreover, the discussions are much more “descriptions” of the results than “discussions”. Thus, I recommend it for publication, but after a major revision.

Major points:

It is very difficult to fully understand the experimental design shown in Figure 4, which is very naive. This reviewer had many troubles in order to comprehend the “actuating unit”, but there is no mention of what the “flexible electrode” is made of. Also, is the power amplifier applying positive (with one electrode grounded), negative (with one electrode grounded) or positive and negative potentials?

What is the software used to analyze the images? ImageJ? MATLAB? Selected subsequent images would fit just fine in Figure 9.

Are the error bars the result of how many independent experiments? In fact, many of the results presented in Figure 9 are within the experimental error. Perhaps, a PU membrane with a larger diameter and digital image correlation data could help the authors to properly investigate the electromechanical coupling in their material.

Some minor points:

1)    Figures must be improved;

2)    All tables could be simplified and/or moved to a Supplementary Material;

3)    The “conclusions” section is more likely a summary of the results than a conclusion;

4)    SEM images before and after the electrostrain tests could revel if there is any deformation hysteresis or even permanent structural modification.

Reviewer 3 Report

The authors proposed a new route in developing dielectric elastomers. Instead of modifying chemical structure of elastomer, it is filled with microcapsules, containing SWCNT. The paper clearly presents the methods and research design, discussion and conclusions are adequate. Minor comments:

1. Line 84 - DNF: Dielectric Nano Filler. Dielectric material is usually non-conducting, while SWCNTs are good conductors.

2. Lines 160-161 - dielectric sensitivity coefficient β, i.e., magnetoelectric sensitivity factor. Magnetoelectric seems to be a mistake.

Round 2

Reviewer 2 Report

Although the authors have addressed some of the questions/comments raised in the previous version, this Reviewer still considers that Figure 4 must be replaced/improved.
